# Generative modeling for RNA splicing code predictions and design

## Abstract

Alternative splicing (AS) of pre-mRNA splicing is a highly regulated process with diverse phenotypic effects ranging from changes in AS across tissues to numerous diseases. The ability to predict or manipulate AS has therefore been a long time goal in the RNA field with applications ranging from identifying novel regulatory mechanisms to designing therapeutic targets. Here we take advantage of generative model architectures to address both the prediction and design of RNA splicing condition-specific outcome. First, we construct a predictive model, TrASPr, which combines multiple transformers along with side information to predict splicing in a tissue specific manner. Then, we exploit TrASPr as on Oracle to produce labeled data for a Bayesian Optimization (BO) algorithm with a custom loss function for RNA splicing outcome design. We demonstrate TrASPr significantly outperforms recently published models and that it can identify relevant regulatory features which are also captured by the BO generative process.

## 1 Introduction

Alternative splicing (AS) occurs when multiple unique mRNA isoforms are produced from a single gene, each including or excluding different pre-mRNA exonic or intronic segments. AS greatly increases transcriptome complexity such that a single gene can encode many mRNA isoforms, each of which include a different subset of pre-mRNA segments. Over 90% of human genes undergo AS, with a conservative estimate that at least 35% of human genes switch their dominant isoform across 16 adult tissues [32, 44, 16]. Changes in the produced isoforms can have significant phenotypic effects: Defects in splicing have been associated with numerous diseases [38] while at the molecular level, AS has been shown to change protein function by, for example, removing a nuclear localization signal, affecting an RNA or DNA binding domain of the encoded protein, or regulating gene expression by introducing a poison exon that leads to nonsense mediated decay (NMD) [40, 28].

Following the discovery of RNA splicing in 1977 [2, 7], decades of work has identified hundreds of RNA Binding Proteins (RBPs) that regulate splicing outcome. These RBPs have been shown to bind exons and proximal introns, typically up to a few hundred bases away from proximal exons, to regulate splicing in a condition specific manner [12]. Consequently, in an influential 2008 review article, Chris Burge and Zefeng Wang set a long-term goal for the RNA community to construct a predictive 'splicing code' that will be able, given a genomic sequence and cellular condition, predict the splicing outcome [45]. Splicing outcomes are typically measured as the ratio of isoforms that include or exclude a specific RNA segment (e.g., an exon). This ratio is commonly referred to as 'percent spliced in' (PSI, or $\Psi \in [0, 1]$), and changes in splicing between cellular conditions or due to genetic mutations are expressed as dPSI ($\Delta\Psi \in [-1, 1]$). In this work, we consider two tasks related to splicing: splicing prediction, and splicing sequence design.

Submitted to 37th Conference on Neural Information Processing Systems (NeurIPS 2023). Do not distribute.

**Splicing prediction.** Software to predict splicing was first introduced in 2010, formalizing splicing codes as a supervised learning problem for exon $e$ with differential inclusion/exclusion/no-change in a specific cellular condition (e.g., tissue) $c$. Subsequent work defines the learning tasks as a prediction task for $\Psi_{e,c}$ or $\Delta\Psi_{e,c,c'}$ using a variety of modeling approaches, including Bayesian NNs, autoencoders, MLPs, CNNs, and RNNs [48, 6, 49, 25]. Importantly for the work described here, the best performing tissue specific splicing prediction models to date use hand crafted expert derived regulatory features from the genomic sequence of interest, such as RBP binding based on sequence motifs, secondary structure, and conservation values [21]. Subsequent models used the genomic sequence directly as input (*e.g.*, CNN models) but mostly focused on predicting the effect of genetic mutations [6, 50] with only moderate success in tissue specific predictions.

**Sequence design.** Sequence design for RNA splicing is a new task, similar to ones involving the design of untranslated regions (UTR) in mRNA vaccines for optimal expression [5, 26] or the design of alternative polyadenylation [3]. Similar to the latter, we formulate RNA splicing design as a constraint optimization problem, where we are required to generate a genomic sequence that would have specific splicing outcome characteristics, such as increased exon inclusion in brain. In addition, the generated sequence can also be constrained such that it involves for example no more than $M$ mutations in $N$ locations compared to the given starting sequence. Applications of such a design task can vary from optimizing therapeutics for correcting splicing defects to synthetic biology.

This work offers several contributions. First, we propose TrASPr, a new multi-Transformer based splicing code model, demonstrating it can achieve state of the art results for tissue-specific splicing prediction. Second, we formulate RNA splicing design as an optimization problem involving a deep generative model such that Bayesian Optimization (BO) techniques can be utilized for it. Our BO algorithm for splicing (BOS), uses TrASPr as an Oracle to optimize a VAE under sequence and splicing outcome constraints. We first test TrASPr on RNA splicing data from both mouse and human tissues, demonstrating it achieves state-of-the-art prediction accuracy. Then we show TrASPr detects condition specific regulatory elements using ENCODE data involving three RBP Knockdown (KD) in two human cell lines, and data for tissue-specific regulatory elements from a mini-gene reporter assay. Finally, we demonstrate BOS can effectively mutate a given sequence under a limited number of mutations to achieve a pre-defined tissue specific splicing outcome.

## 2  Background

### 2.1  Quantifying AS events

Splicing codes require training data in the form of quantified AS events across diverse conditions. Such AS quantification nowadays is mostly derived from Illumina RNA sequencing reads. Each experiments includes millions of these ∼100bp long reads that are mapped back to the genome using dedicated tools (*e.g.*, STAR). Dedicated splicing analysis algorithms are then used to first detect the AS events, typically from reads spanning across RNA segments, then quantify those in terms of $\Psi$ or $\Delta\Psi$ as described above. Here we applied the commonly used MAJIQ algorithm [42] to quantify AS as it has been shown to compare well to other tools [27] and carries several additional benefit important for the task at hand. Specifically, MAJIQ allows for the detection of unannotated splice site, splice junctions, exons, and intron retention events. Furthermore, MAJIQ can capture complex AS events involving multiple alternative splice junctions. These characteristics are key for creating a high-quality train and test samples where such variations are controlled (see details below). Specifically, we only

### 2.2  Transformer modeling of RNA sequence

In this work we adapt BERT [9] model to RNA sequences. BERT is a bi-directional transformer-based model, which learns contextual relations of tokens in a text [43]. The BERT model can be pre-trained on large unlabeled datasets of tokenized text using masked token prediction. Here we considered different tokenizing strategies of RNA sequences which are composed of 4 types of ribonucleotide bases ('A','C','G','U'). We settled on overlapping k-mers of length 6 such that the sequence "AUUGGCU" is represented by a string containing two tokens, AUUGGC and UUGGCU. During pre-training all k-mers that include a specific nucleotide are masked as in for example the DNABERT model [22]. However, we found the DNABERT architecture to be unstable and opted to

pre-train a lighter BERT model with only six layers as describe below. In addition to all possible 6-mer combinations of ribonucleotide bases, we include 5 special tokens to represent classification ([CLS]), padding ([PAD]), separation ([SEP]), mask ([MASK]) and unknown ([UNK]). Finally, we extend the vocabulary with additional tokens to capture additional features and information such as the tissue type, species and length tokens.

## 2.3 Notation

We measure splicing across $c \in [1, \ldots, C]$ conditions for events $e \in [1, \ldots, E]$. Each AS event $e$ has a sequence $S_e$ comprised of 4 different regions, each centered around the respective splice site $S_e = \{S_e^1, S_e^2, S_e^3, S_e^4, \}$. Similarly each event has a set of features associated with it such as exon length, conservation etc. denoted $F_e$. Splicing quantification for event $e$ in condition $c$ is denoted $\Psi_{e,c} \in [0, 1]$ and differential splicing as $\Delta\Psi_{e,c,c'} \in [-1, 1]$ accordingly. However we frequently drop the event $e$ or condition $c$ index for brevity.

## 3 Related work

The first splicing code model used boosted decision trees, learned from over 1000 putative regulatory featured derived from the literature [1]. While that first model had only ∼3700 exon skipping AS events to learn from, subsequent models took advantage of more samples from RNA-Seq data that had became available to train Bayesian and deep learning models [48, 47, 4, 6, 50]. The best performance on tissue specific splicing prediction was achived in [21] using a similar set of pre-defined regulatory features that were first condensed using an AutoEncoder, then combined in a MLP. Subsequent works aimed to learn a code directly from the genomic sequence using a variety of architectures. MT-Splice for example used a CNN based architecture with 64 length-9 filters while the more recent Pangolin [50] employed a ResNet architecture originally introduced in the SpliceAI model for detecting cryptic splice site [20]. Both MT-Splice and Pangolin focused on predicting mutations that affect splicing outcome and reported moderate accuracy for tissue-specific splicing prediction.

The RNA splicing design task is new and possibly the only directly related work is Deep Exploration Networks (DEN) by [29]. DEN involves a VAE which generates genomic sequence, the generated sequence is then evaluated by a prediction model for the desired task (*e.g.*, splicing outcome $\Psi$) which is combined, via a hyper parameter $\lambda$, with a second target function that penalizes generated sequences too close to previously generated ones. While similar in spirit, DEN is quite different than the work presented here. First, DEN models the genomic sequence as one long position weight matrix (PWM) that is later collapsed into a specific sequence. The VAE itself is based on a feed forward network and the prediction models are either a CNN or a linear model of k-mer counts as in [34]. The splicing task in that work is also different, involving alternative 5' splice site selection with two relatively short regions downstream of each 5' splice site. Finally, the data used for training and testing the DEN for the above task is distinctly different, based on a large pool of approximately 13,000 synthetic sequences tested in cell lines.

## 4 Data

To pretrain the basic BERT RNA model described above, we extract 1.5 million sequences around splice sites from the GENCODE human pre-mRNA transcripts database. Each sequence was cut to be 400 bases long and centered around the splice site. These sequences are then converted into 6-mers tokens and fed as input to the BERT model.

Similar to previous work, we focus on predicting the inclusion levels of cassette exons. To evaluate performance we use two main datasets. The first is from the mouse genome project (MGP)[23] and involves six mouse tissues (Heart, Spleen, Thymus, Lung, Liver and Hippocampus) with 4-6 replicates each. We also used the same train/test data split used in [21] so that the results can be compared directly to their model. The second dataset is GTEx[8] from which we select six representative tissues/conditions: Heart (Atrial Appendage), Brain (Cerebellum), Lung, Liver, Spleen, and EBV transformed lymphocytes. Note that some conditions are shared between the datasets. This ensures that our model sees sequences from different species but similar tissues. For all tissues and tissue pairs in these datasets, we processed the RNA-Seq using MAJIQ (see Section 2.1) to detect cassette

events with high-confidence quantification for their $\Psi_{e,c}, \Delta\Psi e, c, c'$. In total, we collect E=11346 and E=18278 events from the MGP and GTEx datasets.

**Test set filtering.** The high number of events in our data is partially due to the fact the cassette exons extracted from MAJIQ's splice graphs may be overlapping (*e.g.*, different splice sites used to define the skipped exon). This may be useful for training on diverse exon/intron definitions but care must be taken to avoid information leakage to the test data. This is especially important for large models that can easily memorize genomic sequences [36]. We handle this issue in two ways. First, we fully hide two chromosomes (8, 14 for GTEx and 4, 11 for MGP) for testing, and discard test exons that are too similar to training exons. Sequence similarity was assessed using BLAT [24] with filters for percent identify, difference in length, and the estimated similarity p-value. We consider two filter settings. First, we denote a set of 'Permissive' filters as used in [21], These settings included `maxLenDiff=5`, `minPval=0.0001` and `minIdentity=95`. Because we are using significantly more complex models, we introduce a second set of filters we denote 'Strict' with `maxLenDiff=100`, `minPval=0.001` and `minIdentity=80`. This accounts for short exons with high similarity but that diverge enough relative to their short length to not achieve a significant p-value.

**Test data for mutations and knockdown analysis.** To evaluate the capability of TrASPr and BOS to predict or suggest mutations, we curated two other sets of experimental data. The first one is the RBP Knockdown (KD) experiments from ENCODE [19]. ENCODE data involves two types of cell lines (K562, HepG2) in which various RBP were knocked down, followed by RNA-Seq experiments to measure the KD effect on the transcriptome. Since the ENCODE RNA-Seq data has been shown to exhibit strong batch effects we first performed batch correction using MOCCASIN [39]. Here, we focused on three well studied RBPs (TIA1, PTBP1, QKI) for which there is relatively better sequence motif definitions (*i.e.*, which sequences these RBP are likely to bind) and better experimental binding assays (eCLIP) which indicate regions where these RBPs were found to bind the RNA sequences. To assess whether the splicing code is learning direct regulation by these RBPs we searched for occurrences of these RBPs sequence motifs. Then we filtered those motif locations to be in AS events which had those in the intronic regions proximal to the alternative exon. We furthered filtered those for AS events that had eCLP binding peaks for those RBPs and that their inclusion level was indeed affected upon the RBP KD experiment ($|\Delta\Psi| > 0.15$)). This set of AS events served as putative targets of the above RBPs. We then 'removed' the effect of these RBPs on the set of AS targets by randomly mutating the identified binding motifs. We repeated this process 5 times with different random mutations and the prediction results where then averaged and compared to the wild type (WT) sequence prediction. These *in-silico* predictions of RBPs effects where then compared to those observed in the actual KD experiment. Finally, we also included experiments from a mini-gene reporter assay where the effect of mutating several regions upstream of exon 16 of the mouse Daam1 gene where tested [1].

# 5 Methods

Our method involves three main components depicted in Fig. 1: An elaborate data processing pipeline discussed above, a transformer based splicing prediction model (TrASPr), and a Bayesian Optimization algorithm (BOS) to design RNA with desired properties. We now turn to describe the two latter modeling components in order.

## 5.1 TrASPr

### 5.1.1 Pre-training RNA splice site BERT model

The foundation model for TrASPr is a 6 layer BERT model which is pretrained on human RNA splice sites (Fig. 1b). Following the pretraining step, as in [22], TrASPrtakes an RNA sequence converted to 6-mer tokens as input, but instead of using the BERT default max length, we feed the model with 400 bases long sequences where the splice site (either 5' or 3' splice site, as shown in the cartoon) is in the center.

For pre-training, we follow BERT in randomly choosing 15% of tokens, but additionally mask the surrounding k tokens for each one to account for our overlapping 6-mer tokenization. We used standard masked autoencoding training, calculating the loss from the original 15% of tokens that

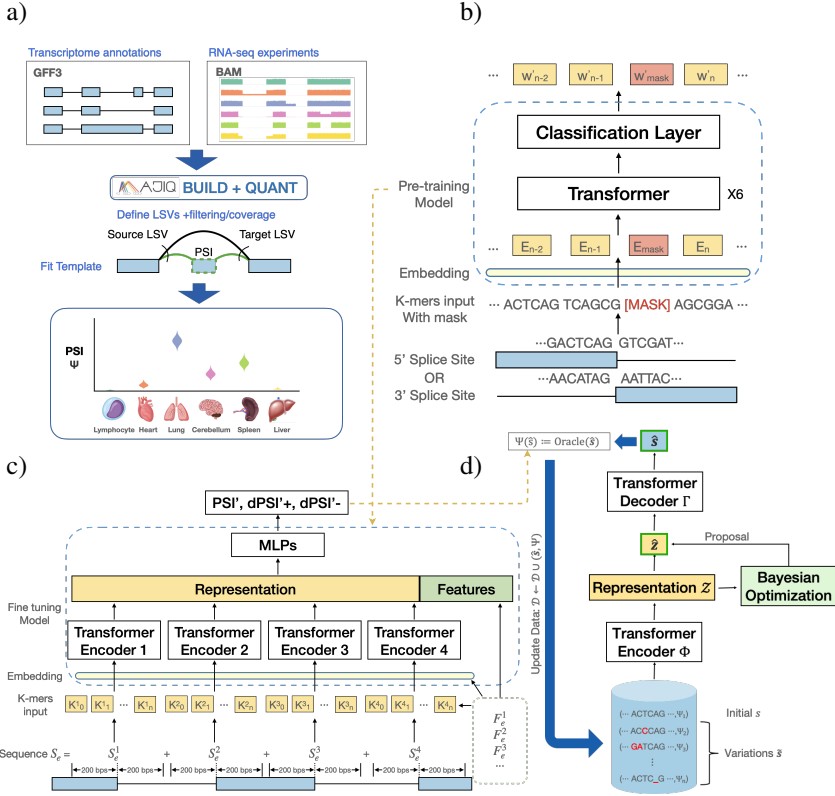

Figure 1: Pipeline and structures of our model. a) Data curation pipeline. b) Pre-training stage. c) Fine-tuning TrASPr. d) BOS structure and flow.

were masked. We pretrain for 110k steps with a batch size of 40. The learning rate was set to 4e-4 and we used a linear scheduler with 10k warm-up steps.

### 5.1.2 The TrASPr model and fine-tuning

Here, we describe finetuning our TrASPr model from the pretrained model described above, as depicted in Fig 1c. For any given AS event $e$, the input to TrASPr is a sequence composed of four sequences $S_e = \{S_e^i\}_{i=1}^4$ such that each $S_e^i$ covers the exonic and intronic regions surrounding one of the four splice sites involved in the exon skipping AS event $e$. Each $S_e^i$ is fed through a matching pre-trained transformer $T^i$, which also accepts additional event features $F_e = \{F_{e,i}\}$ (see below). The latent space representation from each transformer $T^i$, captured by their respective CLS tokens, are concatenated together along with the feature set $F_e$ and fed into a 2 hidden layer MLP with width 3080 and 768.

**Event features.** The additional feature set $F_e$ includes the exon and intron length information as binned tokens, as well as the tissue type. We additionally include conservation values generated based on PhastCons score[37] for each k-mer in the sequence. Exons generally have significantly higher conservation values, as these reflect selection pressure due to non splicing related function (coding for proteins). We therefore used the mean of all conservation scores to fill the exon regions but kept the original scores for the introns.

**Supervision.** We follow [21] and define targets based on measuring both splicing outcomes and *changes* in splicing outcome for an event $e$ in two different conditions $c, c'$. Specifically, the target variables included:

$$T_{\Psi_{e,c}} = E[\Psi_{e,c}], \ T_{\Delta\Psi+_{c,c'}} = |\max(\epsilon, E[\Delta\Psi_{c,c'}])|, \ T_{\Delta\Psi-_{c,c'}} = |\min(\epsilon, E[\Delta\Psi_{c,c'}])|$$

Here $E[\Psi_{e,c}], E[\Delta\Psi_{c,c'}]$ represent the posterior expected values for PSI and dPSI as estimated by MAJIQ from the RNA-Seq experiments [42]. The $T_{\Delta\Psi+_{c,c'}}$ target captures events with increased

inclusion level between tissue c and c' while $T_{\Delta\Psi-c,c'}$ captures events with increased exclusion, forcing the model to focus its attention on those. To avoid gradient issue, we use random small number between 0.001 and 0.002 as $\epsilon$. For all of those target variables we use the cross-entropy loss function which performed better than regression. In the fine-tuning step, we train the model with 2e-5 learning rate and batch size of 32 for 10 epochs.

## 5.2 Sequence design for splicing outcomes.

Beyond supervised learning, we also demonstrate that TrASPr can be leveraged to solve sequence design problems. Given a sequence $S_e = (s_1, ..., s_n)$, TrASPr measures the probability that the splice site in the center of $S_e$ is included in some tissue $c$, $\Psi_c(S_e)$. This value can directly be used as the basis for optimization problems, where we seek to design new sequences $\tilde{S}_e$ that differ from $S_e$ only slightly, but exhibit altered splicing outcomes. Formally, we define these optimization problems:

$$\arg\min_{\tilde{S}_e} \Psi_c(\tilde{S}_e) \text{ s.t. } \text{lev}(\tilde{S}_e, S_e) \leq \tau \text{ or } \arg\max_{\tilde{S}_e} \Psi_c(\tilde{S}_e) \text{ s.t. } \text{lev}(\tilde{S}_e, S_e) \leq \tau \qquad (1)$$

Here, $\text{lev}(\tilde{S}_e, S_e)$ denotes the Levenshtein distance between $\tilde{S}_e$ and $S_e$. Solving the minimization problem is equivalent to finding a small perturbation (up to edit distance $\tau$) of $S_e$ that *reduces* inclusion in the target tissue $c$ by as much as possible. The maximization problem corresponds to *increasing* inclusion. In practice, we add additional constraints that $\forall c' \neq c$, $\Psi_{c'}(\tilde{S}_e)$ cannot be reduced below 0.05. This additional constraint prevents an optimization routine from destroying splicing to such an extent that all inclusion levels are driven to zero.

To solve these optimization problems, we adapt recent work in latent space Bayesian optimization (LSBO) for black-box optimization problems over structured and discrete inputs [30, 41, 14, 31, 46, 35, 15, 17, 18]. LSBO solves structured optimization problems using two primary components: (1) a deep autoencoder (VAE) model, and (2) a Bayesian optimization routine.

**Variational autoencoders for LSBO.** In LSBO, we train a DAE that assists in reducing the discrete optimization problem over sequences $\mathcal{S}$ to a continuous optimization problem over the *latent space* of the VAE, $\tilde{\mathcal{Z}} \subset \mathbb{R}^d$. Leveraging the same data used to train TrASPr, we train a 6 layer Transformer encoder $\Phi : \mathcal{S} \rightarrow \mathcal{P}(\mathcal{Z})$ and 6 layer Transformer *decoder* $\Gamma : \mathcal{Z} \rightarrow \mathcal{P}(\mathcal{S})$ [43]. The encoder $\Phi(S_e)$ maps sequences $S_e$ onto a distribution over real-valued, continuous latent vectors $\mathbf{z}$. The decoder $\Gamma(\mathbf{z})$ (probabilistically) reverses this process. The parameters of $\Phi$ and $\Gamma$ are trained so that roughly we have $\Gamma(\Phi(S_e)) \approx \tilde{S}_e$. Because we only care ultimately about the output sequence $\tilde{S}_e$, here we abuse notation and denote the most probable sequence output from the decoder as $\Gamma(\mathbf{z})$. For optimization, the advantage the VAE provides is the ability to optimize over *latent vectors* $\mathbf{z}$ rather than directly over sequences $S_e$. This is because, for any $\mathbf{z}$ proposed by an optimization algorithm, we can evaluate $\Psi_c(\Gamma(\mathbf{z}))$. We therefore search for a $\tilde{\mathbf{z}}$ such that $\tilde{S}_e := \Gamma(\tilde{\mathbf{z}})$ is a high quality solution to the optimization problem.

**Bayesian optimization.** With the optimization problems in Equation 1 reduced to continuous problems over $\tilde{\mathbf{z}} \in \mathcal{Z}$, we can now apply standard continuous black-box optimization algorithms. Bayesian optimization [13] is among the most well studied of these approaches in the machine learning literature. In iteration $n$ of Bayesian optimization, we have a dataset $\mathcal{D}_n = \{(\mathbf{z}_i, y_i)\}_{i=1}^n$ for which $y_i = \Psi_c(\Gamma(\mathbf{z}_i))$ is the known objective value. We train a surrogate model of the objective function using this data–most commonly a Gaussian process [33]–and use this surrogate to inform a policy–commonly called an *acquisition function*–that determines what latent vectors $\mathbf{z}_{n+1}$ to consider next. In this paper, we use LOL-BO [30] as our base, off-the-shelf LS-BO algorithm. To accommodate the constraints in Equation 1, we modify LOL-BO to utilize SCBO [11] rather than TuRBO [10] as the underlying optimization routine. As with the objective, the Levenshtein constraint is evaluated on decoded latent vectors: $\text{lev}_{\mathcal{Z}}(\mathbf{z}, \mathbf{z}') = \text{lev}(\Gamma(\mathbf{z}), \Gamma(\mathbf{z}'))$.

## 6 Results

In this section, we compare TrASPr with state-of-art methods on predicting condition specific splicing changes, assess its ability to predict the effect of changes in *trans* (RBP KD) or *cis* (mutations in a mini-gene reporter assay) using *in-silico*, then assess the ability of our proposed generative algorithm BOS to propose sensible sequences for a user defined splicing outcome.

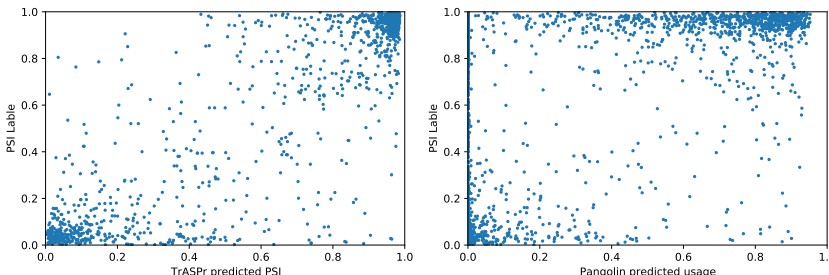

Figure 2: Comparison of PSI prediction results on GTEx dataset. Scatterplots show the prediction vs. RNA-Seq values for TrASPr (left, pearson 0.81) and Pangolin (right, pearson 0.173).

## 6.1 Predicting exon inclusion levels across tissues

We first evaluate TrASPr on the task of predicting $\Psi$ using human GTEx data, comparing to Pangolin. Pangolin uses the SpliceAI model architecture [20] and was originally trained on data from four species, each with four tissues. Pangolin model is unable to define specific splicing events such as cassette exons. Instead it uses a 10Kb sequence window and predicts a 'splice usage' value for the position in the center, constructing a separate model for each tissue. To make Pangolin comparable, we feed the 3' and 5' splice site of each alternative exon $e$, then calculate the average of these two. Performance was evaluated on shared tissues and test chromosomes as used in [50]. Our model achieved significantly higher Pearson correlation for PSI prediction (0.81 vs 0.17 see Fig 2), even though the training set is smaller due to only using overlapping tissues. Taking a closer look both models work well on most of low PSI cases. However, Pangolin performance suffered on some high inclusion cases, assigning low inclusion values. This result might be because of condition specific regulation, because the relevant sequence context is outside the 10kb fixed window used by Pangolin, or because other splicing signals in that window 'confused' the model with respect to quantifying the inclusion of the cassette exon. We note that as mentioned by the authors in [50], predictions for tissue specific splicing changes were not very accurate and we therefore not include them here.

Next we turned to assess tissue specific differential splicing predictions. We compared TrASPr against a previous model that used the same target function but employed manually curated features parsed through an AutoEncodeer and several layers of MLP (denoted 'AE+MLP feature model') [21]. This curated feature set was only available for the MGP dataset and so we assessed performance on this data using the same train and test set definitions as by the authors. Fig 1 and Table 1 show TrASPr significantly outperformed the AE+MLP model in identifying both differentially included and differentially excluded events, especially in terms of AUPRC (every pair of tissues is a point in the scatter plot with blue crosses and brown minuses each denoting evaluation on a set of differentially included or excluded events respectively). However, when we applied a more stringent filtering criteria on the test set TrASPr performance degraded while, surprisingly, AE+MLP performance improved. The degraded performance of TrASPr may be due to the fact the model was able to relate mouse and human AS events that are somewhat similar, but the fact performance for AE+MLP model improved may point to some specific characteristics of the stricter dataset that may have made it easier to predict using the pre-defined feature set.

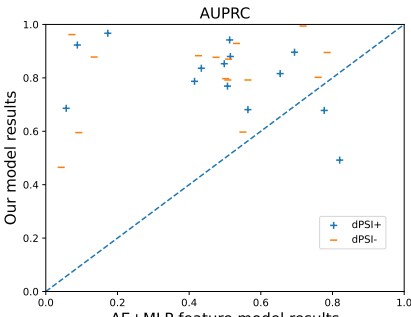

| | AE+MLP feature Model | | TrASPr | |
|---|---|---|---|---|
| Filter | Default | | | |
| AUPRC | 0.4861 | 0.4438 | 0.6079 | 0.6038 |
| Spearman | 0.5503 | | 0.6867 | |
| AUROC | 0.8712 | 0.8502 | 0.8895 | 0.8892 |
| Filter | Strict | | | |
| AUPRC | 0.5388 | 0.4874 | 0.5579 | 0.5176 |
| Spearman | 0.5962 | | 0.5917 | |
| AUROC | 0.8909 | 0.8766 | 0.8740 | 0.8695 |

Figure 3: Comparison of dPSI prediction results on MGP dataset

Table 1: Results on MGP dataset compared with AE+MLP feature model

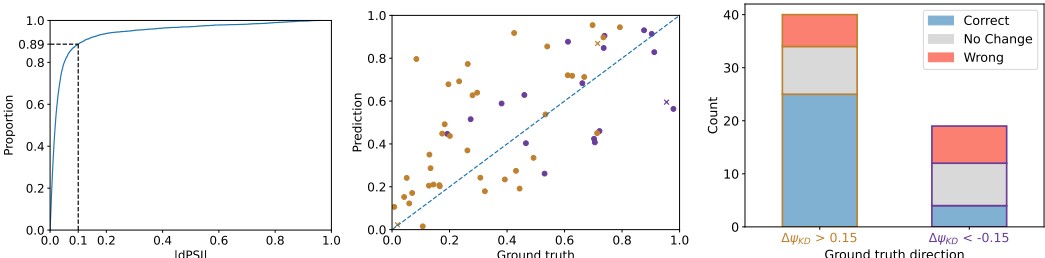

Figure 4: TrASPr prediction results on ENCODE dataset. Figures from left to right: (a) CDF of the difference between TrASPr predicted PSI and the ground truth on wile-type cases from GTEx+ENCODE test set. (b) TrASPr PSI prediction on wild-type AS events compared to RNA-Seq ground truth. Brown and purple indicates AS events whose inclusion are increased/decreased respectively upon RBP KD. (c) TrASPr dPSI direction prediction results for the events in (b). Blue, grey and red color bar means correct, no change, and wrong direction prediction respectively.

## 6.2 Predicting the effect of RBP KD and mutations

We next turned to assess TrASPr ability to predict the effect of RBP KD and mutations. For this we first retrained the model using ENCODE data described in Section 4. First we assessed whether TrASPr is able to accurately predict exon inclusion in those new conditions. As shown in Figure 4a, TrASPr was able to predict $\Psi$ within 10% accuracy in almost 90% of the test cases, indicating the model was able to learn inclusion levels in those cell lines. Next, we focused on the set of putative RBP cassette exons targets shown in Figure 4b, where brown and purple represent events whose inclusion levels went up or down upon KD respectively. We find the WT $\Psi$ predictions for these correlated well with the experimental results (pearson's 0.65), and therefore continued with mutating the specific sequence motifs suspected to be the binding sites for the three RBPs of interest.

Before we could evaluate predictions for *in-silico* mutations we first needed to assess the significance of any given prediction. To achieve this, we randomly mutated sequences in the same set of exons, selected from the same distribution of distances as the original motifs (the distance can greatly affect the null distribution), but made sure non of these randomly chosen regions hit any of the 'real' motifs. We then used the 95 percentile of effects observed in this set as our threshold to call changes. The results of the *in-silico* mutagenesis experiment are summarized in Figure 4c. The left stacked bar shows cases whose PSI increased after RBP KD and the right bar shows decrease PSI cases. The correct(blue) and wrong(red) indicates if the predicted direction is the same as the label and no change(grey) means predicted dPSI was below the 95% cutoff described above. Overall, TrASPr performed well on most of the positive direction cases but predicted around half of negative direction cases as no change. The correlation coefficient for the dPSI effects was 0.34 with an associated p-value of 0.0192. The fraction of correctly called changes was over 50% with a p-value of 0.0001 (TNOM based test).

Finally, we assess TrASPr predictions for mutations introduced in a mini-gene reporter assay around a neural specific exon 16 in the mouse Daam1 gene. Similar to the ENCODE RBP analysis, we find TrASPr correctly predicts the effect of mutations in 7 out of 9 the cases (p-value 0.0012), as shown in Fig 5a. Here too, we find the model correctly predicts increased inclusion but the two mutations decreasing inclusion of exon 16 were not predicted correctly. We note these cases both involved region 11 (marked in red) which the model failed to capture.

## 6.3 Assessing BOS sequence generation

We used TrASPr as an Oracle for our BOS algorithm to generate AS event sequences with edit distances from an original sequence of no more than $\tau = 30$. First, we asked BOS to increase the inclusion levels of lowly included cassette exons from Figure 4b. From the generated 214 sequences with increased inclusion (dPSI>0.2), our BOS algorithm significantly increased PSI(dPSI>0.5) for 46 of them. Most of the mutations were introduced around the relatively weak splice sites surrounding these AS events, which made biological sense. Scanning for the known motifs we found BOS also generated 15 cases where the known RBP regulatory motifs (TIA1, PTBP1 or QKI) were mutated to increase inclusion. When assessing BOS on the daam1 exon 16 we again found many of the mutations increased inclusion by affecting the splice sites as expected (Figure 5b). However, zooming

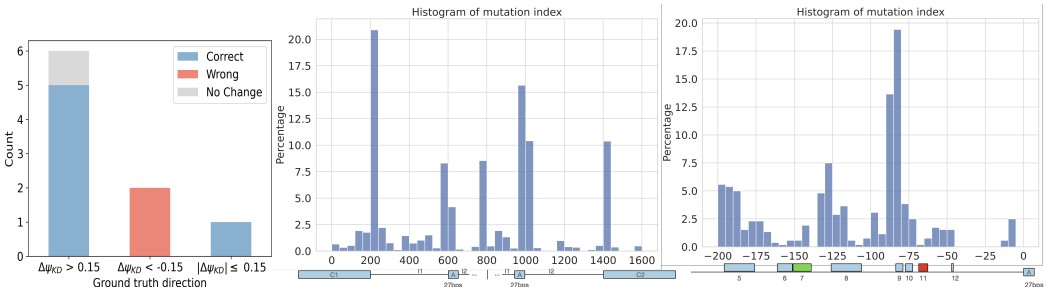

Figure 5: TrASPr dPSI prediction results on Daam1 gene. Figures from left to right: (a) TrASPr dPSI direction prediction results on 9 mutation regions of Daam1 gene. (b) Overall distribution of mutation hits generated by BOS. (c) Distribution of mutation hits among experiment regions.

in on the upstream intron we found BOS frequently mutated the validated regulatory regions avoiding the region of small/little effect (green) and the area that caused decreased inclusion (red).

To assess the efficiency of BOS, we compared its efficacy on *reducing* $\Psi$ for a given sequence with a baseline method which randomly mutated 3, 6, 15 and 30-mers in different regions. We then calcualted how many of these mutations actually changed the PSI by at least 0.2 based on the TrASPr oracle. In the end, the best setting(30-mers) successfully generated 177 out of 4392 sequences(4.03%). BOS generated 12,066 successful sequences(dPSI>0.2) with only ~40k trials(>25%), significantly outperforming the baseline. Overall, these preliminary results indicate that BOS is able to efficiently capture regulatory elements in a given sequence, including both splice site signals as well as deep intronic elements, then capitalize on those to generate sequences matching a given splicing target function.

## 7 Discussion

In this study, we offer two main contributions. First, we propose a new tissue specific splicing code model, TrASPr. TrASPr leverages recent advances in LLMs utilizing Transformer based architecture. The architecture of TrASPr allows it on one hand to benefit from the Transformer attention mechanism while at the same time, by utilizing several Transformers each focused on a specific region, keep the model's attention on areas most relevant for splicing regulation without resorting to extremely large models. We demonstrated TrASPr was able to significantly improve performance in both PSI and dPSI predictions on several datasets compared to previous state of the art. These included CNN based models as well as models utilizing expert derived regulatory features that were fed into a DL model.

The second contribution in this study is in formulating the design of RNA sequences with specific splicing characteristics as a Bayesian Optimization problem. We then proposed the BOS algorithm, which uses TrASPr as an oracle, to solve this design problem with biologically plausible mutations. The RNA design task can be leveraged for synthetic biology studies and for therapeutic design (*e.g.*, which sequence to target with ASO therapy or with prime editing). We showed BOS can effectively propose sequences that exhibit the desired splicing changes, mutating both core splicing signals and intronic regulatory elements.

It is important to keep in mind that the labels used for assessing the prediction tasks presented here are inherently noisy and limited in number. For example, RNA-Seq quantifcations are noisy measurement, as are the RBP binding assays (eCLIP). The RBP regulatory motifs are crude as well. This means many targets might be missed while the changes upon RBP KD can be due to indirect affects (*e.g.*, another RBP affected by the KD) or other sequence motifs. Thus, the work presented here should be viewed more as a proof-of-concept outlining exciting directions for future developments rather than a finished product. Specifically, combining the models we propose with high-throughput mutagensis experiments appears as an exciting direction to explore.

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
