# OpenReview forum: "Generative modeling for RNA splicing code predictions and design"
_NeurIPS.cc/2023/Conference — Submitted to NeurIPS 2023_

### Official Review · Reviewer_Kz6p · 2023-07-01

**Soundness:** 3 good
**Presentation:** 3 good
**Contribution:** 3 good
**Rating:** 6
**Confidence:** 3

**Summary:**

This paper presents a tissue specific transformer based splicing prediction model, TrASPr along with a Bayesian Optimization algorithm, BOS, capable of designing RNA with desired properties. The authors start by demonstrating the performance of TrASPr on RNA splicing data from both mouse and human tissues. Next, the authors assess the model’s ability to detect condition specific regulatory elements using ENCODE data involving three RBP Knockdown (KD) in two human cell lines, and data for tissue-specific regulatory elements from a mini-gene reporter assay. At last, TrASPr is used as an Oracle for BOS to generate AS event sequences with desired properties and an evaluation of BOS performance is presented.

**Strengths:**

[-] Originality; This paper utilizes recent advances in large language models (LLMs) to define a splicing prediction model. While this is not the first attempt to use LLMs for nucleotide encoding, the authors improved existing models and incorporated existing prior knowledge through dedicated features. Furthermore, the authors use the model as an oracle for a generative model for RNA splicing design.

[-] Quality; The paper is well-written and presented. The framework seems well thought through; combining both expert prior knowledge regarding the problems tackled and state-of-the-art computational models.

[-] Clarity; The paper is self-inclusive, presenting the reader with all necessary information from the background regarding the biological problem, its complexity, and motivation regarding its importance. Following that, all framework parts are clearly presented.

[-] Significance; This work provides promising results towards utilizing LLMs for predicting splicing events and further using such these to train an RNA design model. While this work is only the initial step towards obtaining a robust, reliable framework that solves this task it is of great significance as it advances the field.


**Weaknesses:**

[-] Reproducibility code; the authors claim for reproducibility however no code was provided. Providing the code could improve the understanding and evaluation of the presented framework.

[-] Structure; the paper is generally well structured in providing all necessary information however the organization could be improved. For example, the background section already contains model implementation details and prior attempts are split between the introduction and related work. In line with the latter, it would be beneficial if the authors could provide a more elaborate description of previous work, specifically for the prediction task.

[-] Evaluation; the author’s explanation regarding the degradation in performance over the “Strict” test set is not convincing, and following the rationale behind the necessity of the “strict” test set raises some concerns. It would be insightful if the authors could look more into this point, potentially testing on alternative data to obtain a better understanding of this.

[-] minor (these are provided to improve the manuscript’s readability);

[--] incomplete/unreadable sentences; a few sentences in the text are incomplete or unclear (e.g. line 78, lines 256-259)

[--] Space after TrASPr is omitted in many places in the manuscript (probably as a result of latex macro).

[--] Figure 1; Figure 1a is never referenced (and its components are hence not explained), the relationship between 1b-1c could be depicted better (in line with many transformer visualization models).


**Questions:**

1. Could the authors provide a code base for evaluation?
2. Could the authors extend the analysis on differences in performance over the MGP dataset or provide an additional example?
2. Can the authors suggest an alternative baseline, similar to DEN but which can be evaluated in the same setting, to compare the performance of the design class?


**Limitations:**

Yes, limitations have been addressed.

---

> ### Author Rebuttal · Authors · 2023-08-09
>
> Reviewer Kz6p
>
> **Weaknesses**:
>
> ==========
>
> [-] **Reproducibility code**; the authors claim for reproducibility however no code was provided. Providing the code could improve the understanding and evaluation of the presented framework.
>
> **Reply**:
> Agreed! Code will be made available after publication.
>
> [-] **Structure**; the paper is generally well structured in providing all necessary information however the organization could be improved. For example, the background section already contains model implementation details and prior attempts are split between the introduction and related work. In line with the latter, it would be beneficial if the authors could provide a more elaborate description of previous work, specifically for the prediction task.
>
> **Reply**:
> Thank you for the suggestion. The model implementation details will be moved to the model introduction part and prior attempts will be combined together. In terms of previous work, we plan to include more work about RNA splicing prediction with other structures and other biological prediction tasks with BERT related models (see also response to other reviewers).
>
> [-] **Evaluation**; the author’s explanation regarding the degradation in performance over the “Strict” test set is not convincing, and following the rationale behind the necessity of the “strict” test set raises some concerns. It would be insightful if the authors could look more into this point, potentially testing on alternative data to obtain a better understanding of this.
>
> **Reply**:
> We are unsure about the “not convincing” comment. The focus of this work is on the new model, TrASPr, and we clearly show that with a more strict threshold we get somewhat of a drop in performance, which makes sense. We assume the comment refers to the old AE+MLP model compared against. We agree that the result for that model showing improvement was not immediately clear to us either. We don’t have additional data for this model (the model was published with the analyzed dataset included here, any additional data would require creating it based on the original model curated feature set). We think that some explanation could be in the fact that only a subset of splicing patterns are conserved across species (please see the response to reviewer XeDB for more details), combined with the fact the AE_MLP model only used a limited set of pre-computed features. One way we could potentially look into this further is to look into the splicing patterns of the specific samples which were removed between the “strict” and more “permissive” thresholds. Are those similar or different compared to the matching ones in the training set? We plan to add that analysis for the final version.
>
> ===========
>
> [-] **minor** (these are provided to improve the manuscript’s readability);
>
> [--] incomplete/unreadable sentences; a few sentences in the text are incomplete or unclear (e.g. line 78, lines 256-259)
>
> [--] Space after TrASPr is omitted in many places in the manuscript (probably as a result of latex macro).
>
> [--] Figure 1; Figure 1a is never referenced (and its components are hence not explained), the relationship between 1b-1c could be depicted better (in line with many transformer visualization models).
>
> **Reply**:
> Thank you! We will work to address all of the above comments in the revision.
>
> **Questions**:
> Could the authors provide a code base for evaluation?
>
> **Reply**:
> Yes - please see the response above.
>
> **Questions**:
> Could the authors extend the analysis on differences in performance over the MGP dataset or provide an additional example?
>
> **Reply**:
> In the paper, we provided some results on both MGP and the GTEx (human) dataset. In addition to that, we have done another round of ablation study to better understand the importance of each component of the model and inputs (see response to reviewer aUkC). See also comment above about analyzing performance differences in the MGP data. We definitely want to keep introducing additional datasets/conditions including genetic variants but these are unlikely to make it into this current manuscript.
>
> **Questions**:Can the authors suggest an alternative baseline, similar to DEN but which can be evaluated in the same setting, to compare the performance of the design class?
>
> **Reply**:
> Yes - please see above response to reviewer XeDB on question “BO part of paper barely discussed…” for details.

---

> > ### Comment · Reviewer_Kz6p · 2023-08-13
> > **post-rebuttal comments**
> >
> > I would like to thank the for taking the time and responding to my review.
> >
> > While the authors have directly addressed some of my concerns, a majority of the response relies on changes or inclusions that will be provided in a revised version.
> >
> > Unfortunately, this makes it hard for me to evaluate the quality of these, and based on the response and comments from other reviewers, I tend to keep my current evaluation.

---

### Official Review · Reviewer_XeDB · 2023-07-06

**Soundness:** 2 fair
**Presentation:** 2 fair
**Contribution:** 2 fair
**Rating:** 5
**Confidence:** 4

**Summary:**

The authors develop a new framework to predict alternative splicing of RNA. They then deploy it with adaptations and Bayesian Optimization to design new sequences.

**Strengths:**

I find the validation using the RBP KD experiment interesting. It is great that the knowledge of the biological system can be used to inform your computational experiments.

“We repeated this process 5 times with different random mutations and the prediction results where then averaged and compared to the wild type (WT) sequence prediction.“ - Good to do lots of permutations!

Good to try to remove batch effects with ENCODE data, but going to be hard.

Levenshtein distance between designed sequences is good!

Figure 2 comparison to Pangolin is pretty good and convincing


**Weaknesses:**

Table 1 results of rAUPRC and AUROC are confusing. Can the “feature” and “Model” terms be a bit better defined?

Figure 4b is a bit confusing, and I feel like we need a bit more context. Should things be above or below the line? Can we have a legend for the figure as well?

BO part of paper barely discussed, required changes to core algorithm, and not validated–I would remove. Moreover, the baseline algorithm to compare BO (random mutation) is not a sufficient baseline. What about evolutionary strategies? What about Gibbs sampling?


**Questions:**

“...∼100bp long reads that are mapped back to the genome using 70 dedicated tools (e.g., STAR).”- Citation please?

Section 2.2 What are the “...classification 90 ([CLS]), padding ([PAD]), separation ([SEP]), mask ([MASK]) and unknown ([UNK])....” characters, or what do they mean? Perhaps they are discussed more in detail in the paper, but their relative importance and disambiguation should be described when they are introduced

To what extent is there synteny between mouse and human chromosomes on the held out set? For human chromosomes 8 and 14, and mouse 4 and 11, it seems like there will be some overlap in homologous sequences in the training and test sets. How many sequences are then removed during the BLAT step? Are the fitlers strict enough? Should BLAT just be done on exonic regions, because they are more conserved?

Line 280 - Should be Fig 3 and Table 1

Section 6.2 - “... To achieve this, we randomly mutated sequences in the same set of exons, selected from the same distribution of distances as the original motifs…” What does distance refer to here? Linear genome distance? Levenstein distance?

“We then ’removed’ the effect of these RBPs on the set of AS targets by randomly mutating the identified binding motifs.” What is random here? Did you preserve GC content? Could you just permute the bases?

What is figure 5b trying to show? Is the y axis the number of times a mutation shows up during optimization? Also, is this the figure you are referring to in line 322, where you say figure 4b)?


**Limitations:**

It is unclear how much the BO section is needed. With a sufficient predictor, do evolutionary strategies work?

“Specifically, we only…” Background section is not complete…sort of just trails off.

---

> ### Author Rebuttal · Authors · 2023-08-09
>
> Reviewer XeDB
>
> Weaknesses:
> ===========
>
> (1) Table 1 results of rAUPRC and AUROC are confusing. Can the “feature” and “Model” terms be a bit better defined?
>
> Reply:
> Agreed. The “feature Model” terms should be removed there. They simply refer to the fact that the AE+MLP model is based on predefined/manually curated features.
>
> (2) Figure 4b is a bit confusing, and I feel like we need a bit more context. Should things be above or below the line? Can we have a legend for the figure as well?
>
> Reply:
> Again - agreed. Things should be along the 45 deg line i.e. perfect agreement between predicted and measured PSI values. Most observed values tend to reside around 0 or 1, which is a known phenomenon for splicing. We will improve the figure and add a legend in the final version.
>
> (3) BO part of paper barely discussed, required changes to core algorithm, and not validated–I would remove. Moreover, the baseline algorithm to compare BO (random mutation) is not a sufficient baseline. What about evolutionary strategies? What about Gibbs sampling?
>
> Reply:
> We agree the BO presentation and discussion is limited. We will work to improve on that in the final version where we will include additional supplementary material as suggested by other reviewers. That said, we believe that introducing BO in this context is novel and can potentially garner interest from NeurIPS researchers working on design problems but are not aware of RNA related tasks.
>
> The reviewer raises a good question about additional baselines for the design task. First, we note that since we are now introducing this as a design problem (which was not done before) it means, by definition, there are no available baselines to compare against. This means any other baseline we come up with would require creating another algorithm for this task. Specifically, it is not immediately clear to us how Gibbs sampling/MCMC would be applied here. That said, using evolutionary algorithms directly over proposed sequences seems doable and there are existing packages for those that we could try. We weren't able to complete this in the rebuttal week but we plan to include some runs with those in the final version.
>
> Questions:
> ===========
>
> (1) “...∼100bp long reads that are mapped back to the genome using 70 dedicated tools (e.g., STAR).”- Citation please?
>
> Reply:
> Will add.
>
> (2) Section 2.2 What are the “...classification 90 ([CLS]), padding ([PAD]), separation ([SEP]), mask ([MASK]) and unknown ([UNK])....” characters, or what do they mean? Perhaps they are discussed more in detail in the paper, but their relative importance and disambiguation should be described when they are introduced
>
> Reply:
> Good point. This description assumes prior knowledge which is not appropriate - we will make it more clear in the revised text.
>
> (3) To what extent is there synteny between mouse and human chromosomes on the held out set? For human chromosomes 8 and 14, and mouse 4 and 11, it seems like there will be some overlap in homologous sequences in the training and test sets. How many sequences are then removed during the BLAT step? Are the filters strict enough? Should BLAT just be done on exonic regions, because they are more conserved?
>
> Reply:
> It is important to keep in mind that, unlike gene expression, splicing patterns are generally *not* conserved across species but rather across tissues in the same species (see Barbosa Morais et al Science 2012 and Merkin et al Science 2012). Nonetheless, as these papers show, a subset is highly conserved across evolution. Because we were worried about homologous sequences we created a more strict definition of the filters when using both human and mouse data. The amount of samples (i.e. specific AS events observed in specific tissue) removed are 6328.
>
> (4) Line 280 - Should be Fig 3 and Table 1
> Section 6.2 - “... To achieve this, we randomly mutated sequences in the same set of exons, selected from the same distribution of distances as the original motifs…” What does distance refer to here? Linear genome distance? Levenstein distance?
>
> Reply:
> Linear genome distance
>
> (5) “We then ’removed’ the effect of these RBPs on the set of AS targets by randomly mutating the identified binding motifs.” What is random here? Did you preserve GC content? Could you just permute the bases?
>
> Reply:
> Good question. We simply introduced ten random sequences and computed the average over those. We didn’t try to just permute the bases in that region or preserve the GC content. While these are intronic (non coding) and the motifs are far from those of TF the above suggestions make sense and could potentially help get more stable/accurate results - we should try those for the final version.
>
> (6) What is figure 5b trying to show? Is the y axis the number of times a mutation shows up during optimization? Also, is this the figure you are referring to in line 322, where you say figure 4b)?
>
> Reply:
> In Fig 5b the y axis shows how many times an optimization hits the position. This marginal, which is far from uniform, is sensible given what we know about splicing core machinery and positional bias of intronic regulatory elements. In line 322 we are referring to figure 4b as written. The description there is about which exons were used to test BOS, and those exons are the ones shown in Fig 4b to have low inclusion.
>
> Limitations:
> ===========
> It is unclear how much the BO section is needed. With a sufficient predictor, do evolutionary strategies work?
>
> Reply: Please see the above response/discussion.
>
> “Specifically, we only…” Background section is not complete…sort of just trails off.
>
> Reply:
> Thank you, we will fix this.

---

> > ### Comment · Reviewer_XeDB · 2023-08-15
> > **Follow up**
> >
> > Thank you for your comments and continued work. I have read the comments and will keep the same scores.

---

### Official Review · Reviewer_LDfP · 2023-07-08

**Soundness:** 3 good
**Presentation:** 3 good
**Contribution:** 2 fair
**Rating:** 6
**Confidence:** 4

**Summary:**

The authors propose a new machine learning framework called TrASPr, which is a transformer-based architecture with pretrained RNA language models that is tailored for the prediction and design task of RNA alternative splicing. The authors demonstrate that TrASPr accurately predicts tissue-specific `percent spliced in’ (PSI) scores, and the trained model can facilitate RNA design for specific RNA splicing outcomes.

**Strengths:**

- Overall, the authors introduce and explain the problem of alternative splicing, its significance, and the dataset they used to study this problem quite well for a general reader at a machine learning conference.
- The evaluation setting is generally rigorous, as the authors carefully curated the test set to avoid any information leakage.


**Weaknesses:**

- Some additional work and its relationship to this research should be discussed, such as "RNA Alternative Splicing Prediction with Discrete Compositional Energy Network," which also focuses on the prediction of PSI scores in a tissue-specific setting.
- When evaluating the effect of RBP KD and mutations, the authors first identify a set of RBP motif matching sites that might affect alternative splicing and then check if their model can accurately predict those sites through in-silico mutations. However, this measurement only assesses the model's ability to recover positive samples. The authors should also evaluate and present examples of in-silico mutations on non-regulating motif matches and demonstrate their models' predictions. This is important to show that the model is genuinely learning context-dependent sequence features and not just memorizing motif matches.
- While the formulation of the RNA design problem and the utilization of language models for RNA sequences can be considered novel in the field of RNA splicing, similar techniques have been introduced and used in protein sequence design and protein language models before; it would be nice to discuss some of those (e.g., for a review, see https://doi.org/10.1016/j.cels.2021.05.017), perhaps in Related work.


**Questions:**

Questions: from line 83-87, there are some empirical results, such as “we considered different tokenizing strategies of RNA sequences which are composed of 4 types of ribonucleotide bases (‘A’,‘C’,‘G’,‘U’). We settled on overlapping k-mers of length 6,” and “However, we found the DNABERT architecture to be unstable and opted to pre-train a lighter BERT model with only six layers as describe below.” Are there results or analysis supporting these decisions and claims?

Some suggestions:
- It would be helpful to include a subfigure when introducing the concept of alternative splicing, splicing junction, etc. The authors make good efforts in the main text for getting ML audience to be familiar with the concepts, it would be easier if there are figures or illustrations involved.
- Some typos: e.g. line 9 “on Oracle” -> “an Oracle”, line 278 “Autoencodeer” -> “Autoencoder”


**Limitations:**

The authors discussed the limitations of the noisy labels obtained from biological experiments and concluded that this work "should be viewed more as a proof-of-concept outlining exciting directions for future developments rather than a finished product." It would be helpful if the authors could comment on how many datasets exist and are expected to be generated, and whether these limitations would be addressed with more data or more careful model design.

---

> ### Author Rebuttal · Authors · 2023-08-09
>
>  Reviewer LDfP
>
> Weaknesses:
> =========
>
> (1) Some additional work and its relationship to this research should be discussed, such as "RNA Alternative Splicing Prediction with Discrete Compositional Energy Network," which also focuses on the prediction of PSI scores in a tissue-specific setting.
>
> Reply:
> Yes, we will add a reference to the above mentioned work. We note this work by Chan et al is quite different from the one presented here. Chat et al predict whole transcripts, assume the same genomic sequence, and use additional side information about RBP gene expression levels to then predict (sample specific) tissue specific isoforms (termed PSI as well, but the measured entity is different). These characteristics make the work quite different in terms of the task addressed and closer to DARTS (Zhang et al Nat Methods 2019). Nonetheless we agree it adds more general context and are happy to include it in the related work discussion.
>
> (2) When evaluating the effect of RBP KD and mutations, the authors first identify a set of RBP motif matching sites that might affect alternative splicing and then check if their model can accurately predict those sites through in-silico mutations. However, this measurement only assesses the model's ability to recover positive samples. The authors should also evaluate and present examples of in-silico mutations on non-regulating motif matches and demonstrate their models' predictions. This is important to show that the model is genuinely learning context-dependent sequence features and not just memorizing motif matches.
>
> Reply:
> The reviewer raises some important points which we elaborate on below. First, let us clarify the term “positive samples”. We note that in the cases we evaluate the motifs’ occurrences may push inclusion either up (“positive”) or down (“negative”). However, we think the reviewer means here “negative” as in sequence elements which are non-functional (“non-regulating motif”). We very much agree with the need for such an evaluation but note that the only way to test such sequence elements is by direct mutagenesis experiments (wet-lab), which are labor intensive, low-throughput, and well beyond the scope of a NeurIPS paper.
>
> (3) While the formulation of the RNA design problem and the utilization of language models for RNA sequences can be considered novel in the field of RNA splicing, similar techniques have been introduced and used in protein sequence design and protein language models before; it would be nice to discuss some of those (e.g., for a review, see https://doi.org/10.1016/j.cels.2021.05.017), perhaps in Related work.
>
> Reply:
> We agree. While these works/problems are quite different these are definitely worth pointing out for the interested reader and we will add context/citations as suggested in the revised discussion of related work.
>
> Questions:
> =========
> From line 83-87, there are some empirical results, such as “we considered different tokenizing strategies of RNA sequences which are composed of 4 types of ribonucleotide bases (‘A’,‘C’,‘G’,‘U’). We settled on overlapping k-mers of length 6,” and “However, we found the DNABERT architecture to be unstable and opted to pre-train a lighter BERT model with only six layers as described below.” Are there results or analysis supporting these decisions and claims?
>
> Reply:
> Please see details in the answers to reviewer aUkC on question “How important is the pre-training…”
>
> Some suggestions:
> ==============
> (1) It would be helpful to include a subfigure when introducing the concept of alternative splicing, splicing junction, etc. The authors make good efforts in the main text for getting ML audience to be familiar with the concepts, it would be easier if there are figures or illustrations involved.
>
> Reply:
> That’s a good suggestion! We will create a more substantial introduction as a supplementary, with such a figure to make the topic more accessible to new readers.
>
> (2) Some typos: e.g. line 9 “on Oracle” -> “an Oracle”, line 278 “Autoencodeer” -> “Autoencoder”
> Limitations:
> The authors discussed the limitations of the noisy labels obtained from biological experiments and concluded that this work "should be viewed more as a proof-of-concept outlining exciting directions for future developments rather than a finished product." It would be helpful if the authors could comment on how many datasets exist and are expected to be generated, and whether these limitations would be addressed with more data or more careful model design.
>
> Reply:
> In general, we are aware of several datasets and more data being produced - we will add references to such. We also think that there is still room for improvement, modeling wise, which is an active area in several groups across the world, including ours. We will add that point to the discussion as well.

---

> > ### Comment · Reviewer_LDfP · 2023-08-14
> > **Response to rebuttals**
> >
> > The authors have addressed all my questions, and explained those they cannot address (requiring wet lab experiments). By looking at other reviewer’s comments I agree that a lot of the suggested revisions or changes would make the manuscript better. Taking all this into consideration, I wouldn’t change the score or rating; a weak accept would still be my rating for the current revision.

---

### Official Review · Reviewer_aUkC · 2023-07-24

**Soundness:** 1 poor
**Presentation:** 2 fair
**Contribution:** 1 poor
**Rating:** 4
**Confidence:** 3

**Summary:**

The paper tackles two tasks in alternative splicing of pre-mRNA, where multiple unique mRNAs are produced by including different segments. First, the authors proposed a transformer-based splicing prediction model, TrASPr. A 6-layer transformer model is pre-trained with 1.5M pre-mRNA sequences centered in splice sites. TrASPr utilizes multiple pre-trained transformer encoders for the splicing prediction in a tissue-specific manner. Second, the authors used TrASPr as an Oracle to train and evaluate splicing sequence design based on the Bayesian Optimization algorithm. Through the experiment results, they argue that TrASPr significantly outperforms state-of-the-art models and BOS can generate sequences with desired characteristics.

**Strengths:**

-	The authors tackle important problems in RNA biology. In particular, they argue that the sequence design for RNA splicing is a novel task and it can benefit therapeutics for correcting splicing defects and synthetic biology.
-	To tackle the problems, they adopt a couple of machine learning methods that have shown successes in other domains: pre-training and fine-tuning of language models for biological sequences, and latent space Bayesian optimization (LSBO) over structured and discrete inputs.

**Weaknesses:**

Major comments:
- [Originality] While the methods are novel for their first use for RNA alternative splicing, they still seem like mostly direct applications of widely known machine learning methods. For example, pre-training and fine-tuning of language models seem trivial even for the biological sequences. As referenced in the paper, DNABERT proposed a pre-trained language model for DNAs. There are also plenty of previous works that use pre-trained language models for various protein biology tasks.
- [Quality] While the paper includes a couple of experiment results, I do not think they are sufficient to verify the effectiveness of the proposed methods. It lacks strong baseline models for comparison and ablation studies for thoroughly understanding the proposed methods.
- [Significance] The paper does not bring significant and novel ideas that would be valuable to the broader NeurIPS community.
- [Clarity] I don’t think this is the best version of the paper, considering the broad NeurIPS community is not familiar with computational biology. The explanations are not detailed enough to easily understand the problem and significance of the experiment results.

Minor comments:
- Typo in L10: on Oracle -> an Oracle
- Sec 2.1: Incomplete. It suddenly ends with “Specifically, we only.”
- Sec 7: The authors mostly use the Discussion section for summarizing their contributions rather than discussing notable observations, limitations, and future directions. (except for the last paragraph where they discuss the inherent limitation of experiment data)

**Questions:**

-	How well is the pre-training conducted? Can you provide the training curve and evaluation (e.g. perplexity) of the pre-trained model? How much does it differ from the DNABERT model?
-	How do you include conservation values for each k-mer for the Transformer model? Since the conservation values are not considered in the pre-training, the inputs for the Tranformer encoder will be different from the pre-training and possibly incur out-of-distribution problems.
-	Do the Pangolion and TrASPr share the same training data? If not, is it possible to compare the TrASPr with the Pangolion trained with the same training data? It would more clearly show the effect of the proposed methods.
-	How important is the pre-training? Is it possible to use the pre-trained DNABERT instead? How does the model perform with a randomly-initialized Transformer model? How important is the 6-mer tokenization compared to 1-mer tokenization?
-	How important is the Transformer architecture? How does the model perform with other model types?
-	How important are the event features?
-	It seems TrASPr is used as an Oracle for both training and evaluation of the sequence design. Wouldn’t it produce over-optimistic results?
-	How important is the LSBO? Can you provide comparisons with other baseline methods?
-	The explanations of the biological problem and interpretations of the experiment results should be easier and more intuitive to understand. In addition, more detailed and friendly backgrounds should be included as supplementary.

**Limitations:**

The authors adequately addressed that the experiment data are inherently noisy and limited in number, such that this work should be viewed more as a proof-of-concept rather than a finished product.


---Post-Rebuttal Comments---
Overall, I am inclined to believe that incorporating the authors' responses would indeed enhance the manuscript's quality. Consequently, I have adjusted my rating from 3 to 4. Upon reviewing the comments from other reviewers and the authors' clarifications, it seems I'm not the only one who has struggled to understand the authors' contributions and has concerns about the presentation, especially regarding the BO for sequence design. This suggests that substantial revisions are needed beyond the inclusion of additional experimental results. Although the authors' responses have been comprehensive, I could not give a higher score as I respectfully believe resubmission after revision is more appropriate for this manuscript.

---

> ### Author Rebuttal · Authors · 2023-08-09
>
> Reviewer aUkC
>
> Weaknesses:
>
> Major comments:
>
> **Originality**: While the methods are novel for their first use for RNA alternative splicing, they still seem like mostly direct applications of widely known machine learning methods. For example, pre-training and fine-tuning of language models seem trivial even for the biological sequences. As referenced in the paper, DNABERT proposed a pre-trained language model for DNAs. There are also plenty of previous works that use pre-trained language models for various protein biology tasks.
>
> **Reply**:While we would agree our paper does not focus on new ML methodology/theory, we would like to respectfully push back on the above criticism. In our minds, this criticism mostly reflects a failure on our part to explain what is new/original in our work to those who are not familiar with RNA splicing modeling. Let us try to amend this here:
> It is very true that LLMs have been applied in Genomics with the aforementioned DNABERT as an example. However, unlike DNABERT, which is an “off the shelf” BERT model applied to several Genomic datasets, our work goes beyond direct application of an existing model. Previous attempts at modeling condition specific alternative splicing used either precomputed manually derived features (the latest of that being Jha et al 2017, with an AE+MLP) or very large CNNs (e.g. SpliceAI and Pangolin use 10 Kb windows). The first type of models are limited in scope/ability to generalize to any RNASeq/condition. The latter modeling approach ignores any knowledge about the structure of the underlying transcripts making those harder to interpret. Furthermore, even with those large CNN window sizes, those models are unable to fully capture around 25% of the AS events we use here since in human exons are typically short (~147b long) while introns can span many thousands of bases. It was not immediately clear, at least to us, how to model such large scale events using Transformers that are typically limited in their sequence length. We tried “off the shelf '' sparse transformers of several kinds without much success early on, likely due to the fact much of the sequence is irrelevant for splicing decisions. Thus, we came up with what we consider an original solution to the problem: Instead of ignoring annotation as in recent work (SpliceAI, Pangolin etc.) we utilize it to first identify “templates'' of AS events across the transcriptome, then train multiple transformers, each focused around a specific splice site, and combine those (along with additional features) into an MLP. This allowed the overall sequence to grow linearly with the number of junctions involved (J=4 in this case) such tha complexity of the model is 4xL^2 rather than (4L)^2 or much longer if you just take the entire genomic sequence as in SpliceAI/Pangolin. In addition, we made adaptations to standard BO and used the above model as an Oracle to resolve the issue of BO training. While none of this is novel ML we do believe our modeling approach is new and, as we demonstrate, clearly improves over current SOTA.
>
> **Quality**: While the paper includes a couple of experiment results, I do not think they are sufficient to verify the effectiveness of the proposed methods. It lacks strong baseline models for comparison and ablation studies for thoroughly understanding the proposed methods.
>
> **Reply**:We argue that strong baselines are already included in our work: We used the latest model for tissue specific splicing (Pangolin) for any given RNASeq/condition, and we used the best tissue specific model to date which used curated features. Still, the reviewer makes a good point about the lack of ablation studies and we worked to add those (see below).
>
> **Significance**: The paper does not bring significant and novel ideas that would be valuable to the broader NeurIPS community.
>
> **Reply**:Please see the discussion above. In addition to demonstrating how Transformers could be adapted to RNA splicing modeling, we believe that framing RNA splicing design as a BO problem should be of interest to other groups working on either BO or Genomics.
>
> **Clarity**: I don’t think this is the best version of the paper, considering the broad NeurIPS community is not familiar with computational biology. The explanations are not detailed enough to easily understand the problem and significance of the experiment results.
>
> **Reply**:
> We appreciate the candid criticism and will make a serious effort to have both the problem and the contributions more clear to the non (RNA) CompBio audience in the revised version.
>
> ===========
>
> **Minor comments**:
>
> **Sec 7**: The authors mostly use the Discussion section for summarizing their contributions rather than discussing notable observations, limitations, and future directions. (except for the last paragraph where they discuss the inherent limitation of experiment data)
>
> **Reply**:We will work to improve the discussion by pointing out other possible directions for extensions but also the applications of these models. A good example to note, which highlights the applicability of the proposed models to significant biological problems is the very recent Wagner et al Nature Genetics 2023. There, the authors apply two similar splicing models (MMSplice and SpliceAI - we compared against Pangolin which came out later, used SpliceAI architecture and demonstrated improved performance) to predicting the effect of genetic variants in genomic sequences of patients with rare undiagnosed disease. The authors clearly demonstrate improved performance for detecting variations in splicing when variations in RNA occur in non-CAT (clinically accessible tissue) - see Fig5c in their study. This application also serves to demonstrate why the models presented here are not just a theoretical exercise but can have immediate applications if tuned and packaged correctly.
>
> (due to space limitations we upload a separate PDF with additional figure, table and response to the questions)

---

> > ### Comment · Reviewer_aUkC · 2023-08-13
> > **Post-Rebuttal Comments**
> >
> > I appreciate the authors' detailed responses. They have addressed many of my concerns about the paper. It seems like I misunderstood some parts of the manuscript. Some of my follow-up questions are as follows:
> >
> > - In reference to Table 1 in the separate PDF, there appear to be discrepancies between the results presented in that table and those in Table 1 of the main manuscript. Could you kindly elaborate on whether the ablation studies were conducted under different setups? Further elucidation on these ablation studies would be beneficial.
> > - While the authors have elucidated that the utilization of pre-trained DNABERT yielded unsatisfactory results, it would be helpful to have access to the quantitative outcomes of this comparison. Additionally, could you expound on whether, during the fine-tuning phase, the entire Transformer model is retrained alongside the additional MLPs? Would it be more effective and robust to hyperparameters if the pre-trained Transformer were frozen, with only the MLPs undergoing fine-tuning?
> > - As far as I understand, the claimed main reason for the newly trained Transformer model outperforming the pretrained DNABERT is that it is only pretrained from the splice site sequences rather than the entire genome. To support the claim, would it be feasible to demonstrate the shortcomings of TrASPr when pre-trained with an equivalent volume of sequences randomly sampled from the entire genome?
> > - Regarding the evaluation of BOS sequence generation, in L 320-234, the authors stated "From the generated 214 sequences with increased inclusion, our BOS algorithm significantly increased PSI for 46 of them." However, I am still struggling to comprehend how the authors can determine that the generated sequences truly increased PSI, if they did not use the TrASPr as an Oracle to evaluate them.
> >
> > **Overall**, I am inclined to believe that incorporating the authors' responses would indeed enhance the manuscript's quality. Consequently, I have adjusted my rating from 3 to 4. Upon reviewing the comments from other reviewers and the authors' clarifications, it seems I'm not the only one who has struggled to understand the authors' contributions and has concerns about the presentation, especially regarding the BO for sequence design. This suggests that substantial revisions are needed beyond the inclusion of additional experimental results. Although the authors' responses have been comprehensive, I could not give a higher score as I respectfully believe resubmission after revision is more appropriate for this manuscript.

---

> > > ### Author Response · Authors · 2023-08-19
> > > **Reply to Post-Rebuttal Comments**
> > >
> > > Comment:
> > >
> > > I appreciate the authors' detailed responses.... addressed many of my concerns ..... seems like I misunderstood some parts
> > >
> > > Reply:
> > >
> > > We thank the reviewer for their detailed rebuttal response. We are glad the reviewer found we have addressed many of their concerns and clarified some misunderstandings about our work. We address the follow-up questions below.
> > >
> > > Comment:
> > >
> > > In reference to Table 1....appear to be discrepancies between the results....
> > >
> > > Reply:
> > >
> > > We apologize for the confusion regarding the different experiment settings. In prediction of PSI/dPSI for wild type samples, we have two kinds of datasets. The first dataset is for mouse which samples are generated from the mouse genome project (MGP). This data was used in the original AE+MLP paper so we took the same settings and showed the results in Table1. In the ablation study, we used the human dataset from GTEx, which is far more extensive. The main purpose of the ablation study is to further understand the effects of different components of the model and features.
> > >
> > > Comment:
> > >
> > > ....pre-trained DNABERT yielded unsatisfactory results, ...have access to the quantitative outcomes ...during the fine-tuning phase, the entire Transformer model is retrained....only the MLPs undergoing fine-tuning?
> > >
> > > Reply:
> > >
> > > The reviewer listed several interesting questions here. Starting from the end - Yes, the whole model is retrained with a much smaller learning rate. We tested with freezing transformers and only training the MLP part. However, the performance was worse. This is to be expected: The pre-training is only on splice site recognition but what make a region condition specific still needs to be learned. Freezing the Transformer hurts that learning.
> > >
> > > Regarding a (quantitative) comparison to using DNABERT as the underlying Transformer model: Following the reviewer’s specific request we ran such a model for several days. There were several issues with it. First, using the DNABERT published parameter setting completely failed. As we noted before, we found DNABERT to be highly finicky which made it hard to use in the first place. After parameter search we were able to get it to run successfully and the results were as follows:
> > >
> > > AUPRC: [0.05311496 0.04932114]
> > >
> > > Spearman: 0.11426922300496299
> > >
> > > AUROC: [0.71877873 0.72784285]
> > >
> > > We think the very low AUPRC (but reasonable AUC) has to do with the model preferring to give more weight to the majority class during training (non-changing events). This in turn could be due to the fact that the significantly larger model required us to reduce the batch size from 32 to 16. More experiments and hyper parameter exploration would be needed to look into this but regardless we stress pre-training a different BERT was never the main focus of this paper.
> > >
> > > Comment:
> > >
> > > ...the claimed main reason for ...outperforming ...DNABERT is ...pretrained from the splice site sequences....feasible to demonstrate the shortcomings of TrASPr ...randomly sampled from the entire genome?
> > >
> > > Reply:
> > >
> > > We would like to clarify a few things here. First, it’s unclear to us what exactly was the combination of factors that led our pre-trained model to outperform using a pre-trained DNABERT as described above. For splice-site prediction only DNABERT got slightly worse results compared to TrASPr and for the actual tissue specific predictions much worse results (see above). It could be the finicky nature of the model, the reduced batch size, the fact it was trained on irrelevant genomic data. We think asking us to pre-train TrASPr for several weeks on random genomic data just to delve further into this question is an unreasonable request given that this is not a major point in this paper.
> > >
> > > Commet:
> > >
> > > ...L 320-234, the authors stated... I am still struggling to comprehend...they did not use the TrASPr as an Oracle to evaluate them.
> > >
> > > Reply:
> > >
> > > We apologize for failing to make this clear previously. This analysis is based on TrASPr predictions. The point of this analysis is to use TrASPr as an Oracle, combined with BOS, to efficiently find good candidate sequences to achieve the desired splicing change. The part of the assessment not related to TrASPr is the overlap with the known regulatory motifs around these events and the distribution of locations in terms of whether these make biological sense as we describe in the main text.
> > >
> > > In conclusion:
> > >
> > > Standing issues are (a) better writing/explanations (b) testing evolutionary algorithms. Both are addressable and any result from testing those would be new/interesting. Other reviewers indeed raised that concern but suggested taking the BO out while still giving a higher score. We hope our new set of clarifications/details would help increase the reviewer's confidence in our work rather than recommend a rejection. That said, we understand agreement is not always reached and regardless we very much appreciate the reviewer’s informative comments and time spent to ultimately make our work better.

---

### Official Review · Reviewer_wcPZ · 2023-07-25

**Soundness:** 2 fair
**Presentation:** 3 good
**Contribution:** 2 fair
**Rating:** 5
**Confidence:** 3

**Summary:**

The authors propose two approaches to deal with the problems of alternative splicing (AS).  A transformer architecture-based tissue-specific splicing code model, TrASPr, and a Bayesian Optimization algorithm were performed on the latent variable space of VAE to address the design of RNA sequences with specific splicing characteristics.  The architecture is not so novel, but applying LLM to the AS is worth noting.

**Strengths:**

The proposed methods are exciting and computationally novel.  Using BERT with masking to pre-train the model was a nice touch.  Using VAE with Bayesian Optimization is interesting.

**Weaknesses:**

The proposed method is a nice modeling experiment exploring using LLM on a novel application. However, it lacks reliability as a tool to deal with biomedical problems that can be used for biological research.   The evaluation the author provides shows that the method fails.  The authors mention that TrASPr significantly outperforms AE-MLP in a particular situation but also point out that the performance degrades based on the filtering criteria.  There also seems to be a performance difference in BOS sequence generation based on the edit distance, a hyperparameter that lay users would not know how to tune on their particular problem.

Minor comment:
Line 78 is incomplete.


**Questions:**

Which version of MAJIQ did you use?

**Limitations:**

The proposed method has a potential to be a hypothesis generation method.  However, it lacks the biological soundness and it is unclear how to pinpoint the problem when they arise.  Authors are very casual about the evaluation and the problems that arise when the hyper parameters are chosen inappropriately.

---

> ### Author Rebuttal · Authors · 2023-08-09
>
> Regarding MAJIQ version - we used 2.2
>
> Regarding Weaknesses and Limitations listed by reviewer wcPZ:
>
> We are unsure how to interpret the “biological soundness' ' and ‘lack of reliability’ criticism.  We have worked hard to show not only do the model predictions outperform current SOTA but correspond to known biological motifs where we are able to test for those. Furthermore, we believe there is actual utility in “hypothesis generation” by this method - we give one concrete example for rare undiagnosed diseases (see response to reviewer aUkC). Regarding the critique related to the choice of hyper-parameters setting: We’d like to clarify that we don’t view edit distance as a hyperparameter of BOS per se, because its impact on optimization performance is theoretically monotonic: larger edit distances afford the optimizer more freedom to modify the original sequence. To see that performance is monotonic, simply observe that any sequence with up to k edits is of course a valid sequence with up to k+1 edits, and therefore allowing k+1 edits cannot perform worse than allowing k edits. Therefore, since it is common to prefer parsimonious solutions to problems, edit distance should simply be taken as small as possible while achieving whatever the sequence designer believes to be an adequate change in splicing for their own task. The actual definition of this parameter can be the result of constraints/preferences related to the task. For example, if one was to design RNA edits for therapeutic purpose using base editors (e.g. “fixing” a genetic disorder) they may want to limit the number of locations, their positions and sequence compositions. Here we only explored a simple constraint which is the total number of edits. Since we clearly failed to convey this, we will try to make this point more clear in the revised version.

---

### Author Rebuttal · Authors · 2023-08-09

We would like to thank all the reviewers for their valuable comments. In particular, we appreciate the comments on the importance of the problem we address, the improvement in RNA splicing prediction, the experiment design to support our claim, as well as the positive feedback regarding the flow of our paper. Furthermore, we thank all reviewers for pointing out potential weaknesses and unclear elements. To save reviewers time/effort we addressed all questions and concerns of the reviews in separate responses but pointed to shared elements where relevant. We also uploaded the attached PDF with additional results/figures to address specific questions/concerns raised. We hope you will find our response suitable and look forward to any subsequent inspiring discussion.

NOTE: We were unable to address all of reviewer aUkC questions (which also relate to the uploaded figure and table) so we include those in the attached PDF together with the additional figure and table.

Sincerely,
the authors

---

### Decision · Program_Chairs · 2023-09-21

**Decision:**

Reject

**Comment:**

While the reviewers agreed the paper could be a valuable contribution, there was also a consensus the paper would benefit from another round of revision and review.